# Health knowledge and care seeking behaviour in resource-limited settings amidst the COVID-19 pandemic: A qualitative study in Ghana

Farrukh Ishaque Saah[1]*, Hubert Amu[2], Abdul-Aziz Seidu[3,4], Luchuo Engelbert Bain[5]

1 Department of Epidemiology and Biostatistics, School of Public Health, University of Health and Allied Sciences, Hohoe, Ghana, 2 Department of Population and Behavioural Sciences, School of Public Health, University of Health and Allied Sciences, Hohoe, Ghana, 3 Department of Population and Health, University of Cape Coast, Cape Coast, Ghana, 4 College of Public Health, Medical and Veterinary Services, James Cook University, Townsville, Australia, 5 International Institute of Rural Health, College of Social Science, University of Lincoln, Lincoln, United Kingdom

* fsaahpnur14@uhas.edu.gh

**Data Availability Statement:** All relevant data are within the paper.

**Funding:** The author(s) received no specific funding for this work.

## Abstract

### Background

The emergence of a pandemic presents challenges and opportunities for healthcare, health promotion interventions, and overall improvement in healthcare seeking behaviour. This study explored the impact of COVID-19 on health knowledge, lifestyle, and healthcare seeking behaviour among residents of a resource-limited setting in Ghana.

### Methods

This qualitative study adopted an exploratory design to collect data from 20 adult residents in the Cape Coast Metropolis using face-to-face in-depth interviews. Data collected were analysed thematically and statements from participants presented verbatim to illustrate the themes realised.

### Results

Health knowledge has improved due to COVID–19 in terms of access to health information and increased understanding of health issues. There were reductions in risky health-related lifestyles (alcohol intake, sharing of personal items, and consumption of junk foods) while improvements were observed in healthy lifestyles such as regular physical exercise and increased consumption of fruits and vegetables. COVID–19 also positively impacted health seeking behaviour through increased health consciousness and regular check-ups. However, reduced healthcare utilization was prevalent.

### Conclusion

The COVID–19 pandemic has presented a positive cue to action and helped improved health knowledge, lifestyle, and care seeking behaviour although existing health system

**Competing interests:** The authors have declared that no competing interests exist.

constrains and low economic status reduced healthcare utilization. To improve health systems, health-related lifestyles and healthcare seeking behaviour as well as overall health outcomes even after the pandemic wades off, COVID–19 associated conscious and unconscious reforms should be systematically harnessed.

## Background

SARS-CoV-2, also called COVID–19, emerged as a new strain of Coronaviruses after a cluster of patients were identified with pneumonia suspected to be novel coronavirus pneumonia in December 2019 in Wuhan, China [1, 2]. On March 11, 2020, the World Health Organization (WHO) declared COVID-19 as a global pandemic [3]. As of February, 7, 2021, 106.7 million confirmed cases and over 2.3 million deaths were recorded cumulatively worldwide since the start of the pandemic [4]. In Ghana, 70,046 confirmed cases and 449 confirmed deaths were recorded [4].

The increasing burden of COVID-19 has resulted in various international and national level decisions and protocols, shaped largely by initial responses by high-income countries aimed at ending the pandemic [5]. These protocols include physical/social distancing, wearing of face masks, frequent hand washing with soap, stay-at-home/work-at-home, closure of schools, international travel bans, and economic lockdown of non-essential businesses together with isolation of infected persons and quarantining of exposed individuals [5–7]. With the recent vaccine breakthrough, vaccination is expected to soon become a bigger tool to the fight against COVID-19 [8, 9].

The emergence of COVID-19 has deepened the strain on health systems across the globe more especially the already overburdened health systems of resource-limited countries with 90% of countries in five WHO regions experiencing disruptions to their health services [10]. The greatest difficulties are reported by low- and middle-income countries [10]. In fact, poor health system capacity in such countries makes them highly vulnerable to COVID-19 [11]. Many people in sub-Saharan Africa (SSA) for instance, often lack ready access to clean water for regular hand washing, have poor sanitation, limited to no internet connection for home work and schooling, and little or no savings to support loss of income [12]. Again, in SSA including Ghana, many of the healthcare and public health systems are compromised by inadequate equipment for the care of COVID–19 patients like intensive care unit (ICU) beds, bedside oxygen supply, pulse oximeters, ventilators, and personal protective equipment [5].

Healthcare seeking behaviour (HSB) of a population serves as one major determinant of the health status of a country and thus, its socio-economic development [13]. HSB encompasses a people's inaction, procrastination or action undertaken following recognition by themselves of departing from good health or having a particular health problem to finding appropriate remedy to restore health [14]. There is predominantly poor healthcare seeking behaviour among populations in SSA countries like many other low-and middle-income countries [15]. Poor health seeking behaviour is likely to be deepened with the COVID–19 pandemic because delay in seeking care has been identified to contribute to increased morbidity, mortality and worse health outcomes among patients [13, 16].

Although the COVID-19 pandemic has placed many challenges on health systems worldwide, it has also presented opportunities to re-direct resources to many health promotion interventions and activities where lacking. For instance, the Ghana Health Service (GHS) has intensified public health education across the country using various media. However, the impact of the pandemic and its accompanying national prevention protocols and health

education activities have on the health knowledge, behaviours and care seeking behaviour of the population has not been investigated. We thus, explored the effect of the COVID-19 pandemic and health education intervention on the healthcare seeking behaviour of residents in a peri-urban community in Ghana.

### Conceptual framework

We adapted Andersen's Healthcare Utilization Model (HUM) propounded in 1968 [17]. The conceptualization of healthcare utilization by this model acts on the assumption that a person's use of health service is influenced by three key factors, namely, predisposing factors, enabling factors, and the need for care factors [17, 18], incorporating both contextual and individual level predictors [19]. There are three main tenets of the theory. These are predisposing, enabling, and need factors (Fig 1). The tenets adequately explain the various factors influencing health seeking behaviour amidst the COVID-19 pandemic.

The predisposing factors refer to individual level predictors comprising sociodemographic characteristics such as sex, age [20], religion, education, ethnicity, attitude towards health, and

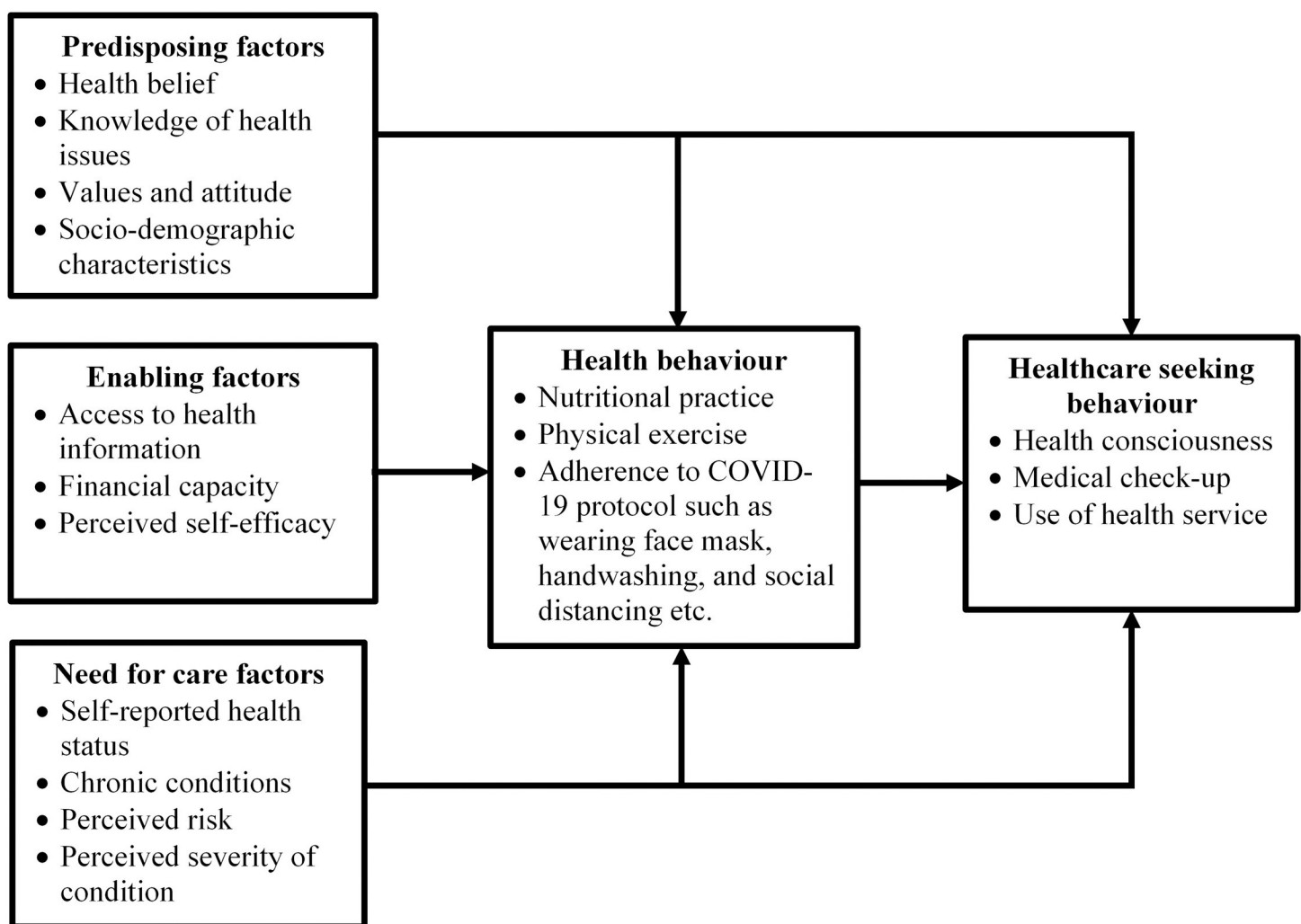

**Fig 1. Health service utilization model.** Source: Adapted from Andersen and Davidson [21].

social relations, health beliefs [18, 19] and contextual factors like social and demographic composition of communities, organisational and collective values, political perspectives and cultural norms. Health knowledge is covered in this tenet as a factor to healthcare behaviours as it shapes beliefs, attitudes, and overall understanding of implications of a specific health behaviour.

According to Andersen and Davidson [20], enabling factors are organisational and financial factors considered to directly affect access to healthcare as well as access to health knowledge and subsequent use of health service [20, 21]. In the context of this study, enabling factors at the individual level include wealth and income at the individual's disposal to cover the cost of assuming positive health-related lifestyles such as good nutrition, physical exercising, wearing face mask and good hand hygiene [19]. They also include travel time to the health facility, the means of transportation, and waiting time for healthcare [21]. In addition, health education and outreach programmes and health policies are factors during this COVID-19 pandemic relevant to individual health behaviour and subsequent healthcare seeking attitude [20, 21].

Furthermore, the need factors refer to individual and contextual level perceptions of the seriousness of a disease or health condition [21, 22]. At the individual level, the model distinguishes between perceived need for health services (how people perceive and experience their own health status (self-rated health), functional state and illness symptoms) and evaluated need (objective measurements of patients' health status and professional assessments, and need for medical care) [20, 21]. Again, contextually, individuals make a differentiation between population health indices such as current COVID-19 infection rate, death rates and overall incidence and prevalence nationally and locally [21]. More so, overall measurements of community health, including epidemiological indicators of COVID-19 morbidity and mortality [22] influence healthcare seeking behaviour.

Despite the few flaws of this model such as disregard for sociocultural dimensions and interactions and omission of social construction of need [23], as well as inadequacy in forestalling service use as predisposing factors might be exogenous and enabling resources are necessary [24], it was considered relevant to this study. This is because its tenets are in line with the study and has the strength of indicating both the micro (individual) and the macro (community) level factors that influence healthcare seeking behaviour. It thus, fits well with the study's objective of assessing impact of the COVID-19 pandemic on the health knowledge and behaviours and subsequent care seeking behaviours of individuals.

## Materials and methods

### Study design

This was an exploratory study adopting qualitative approach. The design allowed for exploring care seeking behaviour by gathering in-depth information through interviews [25]. The design was chosen because it helped to gain in-depth insight on care seeking behaviour amidst COVID-19 with little to no earlier studies to rely upon to predict an outcome for later investigations [26, 27]. It is also flexible and helps address all types of research questions such as the what, why, and how of a phenomenon [25, 26]. The study also used the interpretivist philosophy due to this philosophy's ability to explain how people create and maintain their own social worlds and understanding through personal interpretations of their worlds [28]. The Consolidated Criteria for Reporting Qualitative Research (COREQ) guideline was followed in reporting this study [29].

### Study setting

The study was carried out in the Cape Coast Metropolis of Ghana. The metropolis is one of the 23 administrative districts in the Central Region and its capital, Cape Coast is also the

regional capital. The metropolis lies between longitude 1˚15'W and latitude 5˚06'N with boundaries to the South by the Gulf of Guinea, to the East by Abura Asebu Kwamankese District, to the West by Komenda Edina Eguafo Abrem Municipality, and the North by Twifu Heman Lower Denkyira District. It has a population of 169,894 (7.7% of the region's total population) with 48.7% being males according to the 2010 population and housing census [30]. Also, 90% of the population aged 11 years or older are literate with 67.2% capable of writing and reading English and other Ghanaian languages. Again, 69.5% of the population aged 12 years and above use mobile phones with 32% having access to internet service [30]. The metropolis also has a regional/teaching hospital, and a district hospital among other health facilities including clinics, health centres, and Community Health and Planning Services (CHPS) compounds.

## Study population and sampling

Adult residents in the Cape Coast Metropolis were the study population and were selected using a purposive sampling approach. Only residents aged 18 years and above who had lived in the metropolis for at least six months within the period of COVID-19 pandemic in Ghana were included in the study. The six months inclusion criterion was to ensure that the study included only persons who experienced COVID-19 within the period from April to September 2020 when effects of the COVID-19 pandemic were heavily felt within the Cape Coast Metropolis as COVID-19 case count increased abruptly making the Central Region the third highest region in terms of case prevalence in the country.

The purposive sampling approach allowed us to select only participants who have experienced living in a district with significant COVID-19 cases. Recruitment of participants was done progressively until no new issues were emerging from additional interviewees. Data saturation was achieved after interviewing 20 participants (10 males and 10 females).

## Data collection instruments and procedures

Data were collected face-to-face using an in-depth interview guide. The instrument was self-developed and sectioned into four. While section A collected socio-demographic information of the participants, section B, C, and D, focused on the impact of COVID-19 pandemic on health knowledge, health behaviours, and care seeking behaviours, respectively. The key questions contained in the guide included effects of COVID-19 pandemic on: access to health information, understanding and knowledge of selected health issues, risky health behaviours, adoption of preventive health behaviours like healthy diet and physical exercise, health consciousness, access and use of healthcare services. The questions were generated from literature review and the conceptual framework.

The interviews were conducted at the participants' convenience with support from two trained research assistants who were experienced qualitative researchers with a minimum of bachelor's degree and fluent Fante speakers. Two of the authors who are qualitative research experts, HA and FIS also conducted some of the interviews. The interviews were conducted in Fante, dialect of a major Akan language group predominant in the Cape Coast Metropolis for participants who could not speak or understand English while some were conducted in English. Consequently, the instrument was translated into the Fante language during the two-day training of the research assistants to ensure consistency in translating from English to Fante during interviews. Each interview was between 30–45 minutes and was tape recorded with the consent of the participants. Also, field notes were taken in order to corroborate the transcriptions from the recorded audios.

## Ethical issues

The study obtained approval from the Cape Coast Metropolitan Health Directorate. Prior to inclusion in the study, written informed consent was obtained from the study participants after the study purpose and procedures were explained to them. In effort to protect the study participants and interviewers, the COVID-19 prevention measures of wearing face mask, physically/socially distancing, and using hand sanitizers were ensured. Anonymity was ensured by using pseudonyms combing letters and numbers to identify each participant instead of their personal identifying details. Also, the study data has been stored and password-protected on the personal computer of the corresponding author without access to any third parties to ensure confidentiality.

## Data analyses

All the audio recordings were transcribed and those in Fante and Twi (local dialects of the Akan language) were translated and transcribed into English. While listening to the tapes and using the field notes taken, the transcripts were read and edited to resolve any omissions and mistakes in the original transcripts. Thematic analysis was carried out using NVivo version 10 by first re-reading the transcripts to help familiarize with the data [31]. Initial codes were produced by two of the authors (FIS and HA) from list of ideas found to be interesting and relevant in the data which were later organized into meaningful groups [32]. The generated codes were sorted out and merged to form potential themes [31] based on the research objectives, namely; impact of COVID-19 pandemic on health knowledge, health behaviours and care seeking behaviour; and on literature review and emergent themes. The initial themes were reviewed and refined into final themes taking into consideration internal homogeneity (ensuring everything in a theme is similar) and external heterogeneity (ensuring different contents in different themes) [33]. The themes were defined and named and detailed analysis conducted and written based on how they fit into the broader story of the data. The final step involved full write-up of the report ensuring merit and validity of the analysis using extracts from the data which capture the essence of each theme being demonstrated [31].

## Results

Table 1 presents the themes and sub-themes of the results. Two sub-themes were identified for the impact of COVID-19 on health knowledge whiles there were three sub-themes for the impact of COVID-19 on health-related lifestyles and two for healthcare seeking behaviour.

## Impact of COVID-19 pandemic on health knowledge

The study explored impact of COVID-19 pandemic on health knowledge among participants which found two main positive impacts namely; increased access to health information and improved health knowledge related to chronic diseases, nutrition, hygiene, and risky health behaviours. Regarding increased access to health information, the participants explained that due to the pandemic, many health education activities were ongoing on various media platforms including television and radio stations, community information centres and social media. Mass media, that is, radio and television stations and community information centres, are major platforms used in health education and promotion [34, 35] due to their availability to majority of Ghana's population (69% of women and 80% of men are exposed to radio alone) [36]. This education aimed at not just educating the public on the pandemic but other health issues that were pertinent to transmission of the infection and risk of complications associated with the COVID-19 disease. Also, these education sessions allowed their consumers to participate through text

**Table 1. Themes on the impact of COVID-19 pandemic on health knowledge and care seeking behaviour.**

| Theme | Sub-theme |
|---|---|
| Impact on health knowledge | 1. Access to health information using TV/radio, community public address systems |
| | 2. Improved health knowledge regarding: |
| | • Chronic diseases like hypertension and diabetes |
| | • Risky health-related lifestyles |
| | • Nutritional practices |
| | • Hygiene practices |
| Impact on health-related lifestyles | 1. Stopped/reduced risky behaviour like alcohol intake, sharing of personal items, junk food intake |
| | 2. Started regular physical exercising |
| | 3. Started/increased consumption of fruits and vegetables |
| Effect on healthcare seeking behaviour | 1. Positive effects |
| | • Increased health consciousness |
| | • Started regular check-ups |
| | 2. Negative effects |
| | • Poor health seeking behaviour |
| | • Poor male partner involvement at ANC |

messages and phone calls. For instance, a 36-year-old man said, "*Oh, due to Covid many education activities are ongoing on both tv and radio. . . . and it is helping with getting information like this.*" Another participant, 28-year-old woman noted, "*Now, most of them (television and radio stations) have health education sessions trying to educate the public on the Covid and how to protect ourselves. Even other health issues like hypertension are discussed.*"

Also, a 42-year man stated;

> *For those of us who don't know much about internet, we now get almost all health information we need to prevent health conditions from the regular media like tv and radio. We get to call in to ask questions and they answer us. And the good thing is that it's in Fante.*

Concerning improved health knowledge due to the COVID-19 pandemic, the participants argued that increased access to health information during this pandemic has resulted in better understanding of many health issues. Some also explained that they now understand how to prevent some health conditions like diabetes, hypertension and infectious diseases and how to boost the immune system. Chronic conditions like diabetes and hypertension were noted to dominate many health education programmes due to their increased risk of complications and deaths from COVID-19 with their preventive measures such as healthy diet and increased physical exercise highlighted. Explaining improvement in health knowledge due to COVID-19 pandemic, a 38-year-old woman said, "*I now understand that I need to be careful of types of food I eat, do physical exercises, and not consume alcohol in order to reduce my chances of getting these conditions.*"

A male participant, 29-years old, also noted, "*Now, I know that too much alcohol consumption will likely lead to hypertension and other conditions for life. Initially, I didn't understand why people make a fuss about others drinking alcohol.*" Again, a 47-year-old woman further added;

> *We just eat anything we have at home without regards to whether it improves or destroys our health, we just eat. But now, I know that food is like medicine and when we take lot of fruits and vegetables, we become stronger against diseases.*

## Impact of COVID-19 pandemic on health-related lifestyles

We explored the impact of COVID-19 on the health-related lifestyles among the participants. Three positive impacts of COVID-19 pandemic on lifestyle choices were observed, namely; stopped/reduced risky behaviours, started physical exercising, and started/increased consumption of fruits and vegetables. It was explained by the participants that their increased knowledge and perceived threat of the COVID-19 pandemic have made them assumed positive health-related lifestyles while others have quit risky behaviours such as alcohol intake, sharing of personal items, and consumption of junk foods. The participants explained that they had initiated or increased their consumption of fruits and vegetables and physical exercises as understanding their importance has dominated ongoing health education programmes. Thus, the perceived need to assume this positive health behaviours had improved leading to their decision to apply these health choices. A 40-year-old woman for instance, said; *"Understanding that eating fruits and vegetables helps our body protect against disease in the midst of this Covid, now at least every two days we take banana, oranges and pineapples as part of our dinner."* A 28-year-old man also posited;

> *Oh, as for now, every weekend I and my 'boys boys' go for jogging and sometimes visit the gym to exercise because we understand exercising will keep us healthy and stronger. Our bodies will be strong to fight disease. Previously, aside some community football once a while, I hardly exercised.*

Some of the participants also explained that the pandemic has impacted on their quitting risky behaviours such consumption of junk foods and alcohol. They argued they have stopped or reduced these behaviours due to their improved understanding of risks associated these behaviours and subsequent increased likelihood of complications and death should they be infected with the COVID-19 infection. Explaining this, a 25-year-old woman said;

> *I used not to cook regular meals at home because I mostly buy "indomie" from the fast-food joint in the evening. But I no longer buy that they said such food increases the risk of chronic conditions and subsequent vulnerability to severe complications should I unluckily get Covid. I eat healthier foods now.*

Also, a 38-year-old man explained, *"We're all afraid of being in such conditions (severe COVID-19 illness) due to smoking and alcohol consumption. You know the Covid is everywhere. I have actually reduced my intake of alcohol now."*

## Impact of COVID-19 pandemic on healthcare seeking behaviour

COVID-19 pandemic positively impacted health seeking behaviour among some of the study participants. Positive effect on their healthcare seeking behaviour was explained to be due to the fear of contracting the complicated form of the disease and improved health knowledge. Two aspects of care seeking behaviour were identified to have improved, namely health consciousness and regular medical check-up. Some of the participants posited that they are now more conscious of their health due to improved health knowledge during this period and the need to avoid a health condition taking them by surprise. While some argued that anyone can contract the COVID-19 virus with the rising number of cases as such, they would not want to have the severe form of the disease due to some underlying health conditions, others attributed this to their improved knowledge of health conditions such as diabetes and hypertension. The following quotes summarise their views:

*As for me, now every small thing (physical symptoms) then I feel I need to go to the clinic to check what is the problem. Who knows it could be Covid or some other condition that can worsen my survival should I contract the Covid.*

–Female, 39 years

*I come into contact with many different people while going to work so anything can happen. So, now even small headache or cough then I go to hospital to check what is going on.*

–Male, 43 years

Regarding assuming regular health check-ups as an impact of the COVID-19 pandemic, some participants explained that they now understand the risks of chronic diseases like diabetes and hypertension and that the conditions can be managed well when detected early. As such, they now go for regular blood pressure and blood glucose level checks to help diagnose or ascertain risk of hypertension and diabetes contrary to before the pandemic where these are done only when they were seriously ill. A 44-year-old woman for example noted, *"I have started going for the nurse at the facility to check my BP at least once a month. By that, if I am found to have pressure (hypertension) then they can help me."*

Also, a 51-year-old man said;

*I know I have pressure (hypertension) but normally unless I feel very sick or need to go for new drugs that I go to the hospital–they check me before giving me my monthly drugs. But now, I go to the small clinic here every two weeks to check and even check for diabetes too.*

–Male, 51 years

Furthermore, a significant number of the study participants had negative experiences leading to poor healthcare seeking behaviour as a result of the impact of COVID-19 in terms of negative reception at the facility and poor access to care. Negative reception is explained to mean health professionals applying high-level infection prevention protocols alerted by patients presenting with symptoms such as cough and flu, symptoms also associated with the COVID-19 infection. This the framework considers as health system-related factor to health behaviour. Some participants explained that for fear of being treated like someone infected with COVID-19, they do not go to the health facility with symptoms like cough, flu and fever. For instance, a 33-year-old woman said;

*When I have symptoms like cough or flu, I am unable to go to the hospital because I feel I will be mistakenly treated like someone with the disease (COVID-19). I would rather get some drugs from the drugstore and stay home.*

For others, poor health seeking behaviour is as a result of the fear of coming into contact with COVID-19 infected person or health professional when they visit the health facility. This is due to limited space at the health facilities and sometimes non-adherence to infection prevention protocols such as sanitizing of hands and equipment between patients, increasing risk of direct or indirect contact with the COVID-19 virus. Limited space and risk of nosocomial infection are health system barriers to positive healthcare seeking behaviour as argued in the framework. A 41-year-old man noted;

*What if the person sitting by me at the OPD–sometimes it gets very crowded–has the disease (COVID-19) or the nurse has touched someone who have the disease (COVID-19)? I*

*can get infected so for me unless my condition is serious needing to visit the hospital, I will not go.*

Again, inability to adhere to COVID-19 protocol of facemask wearing was cited to have resulted in poor healthcare seeking behaviour. Some participants argued that one may sometimes forget the face mask at home in the quest of rushing to seek care or may not be able to buy the mask but without the mask, they will not be allowed to enter the health facility to seek care. These barriers, negative health attitude and poor socio-economic status, are considered predisposing factors in the framework. Explaining this, a 38-year-old woman stated;

*Sometimes the only money you have is what to take a car to the hospital and something small to add up for drugs because you are lucky to have (health) insurance. How do you use same money to buy nose mask? But without it you will not be allowed to enter the hospital.*

Also, a 47-year-old woman noted, *". . .sometimes due to how you're feeling because of the sickness, you may be in a rush to go to the clinic. But should you forget the (face) mask, you will be denied entry to the place."*

Additionally, male partner involvement in antenatal care (ANC) was found to be poor due to the impact of the COVID-19 pandemic. A participant explained that male partner involvement at ANC has reduced because of the COVID-19 pandemic citing restriction of male partners by nurses and midwives due to limited space at the facility. Limited space at ANC is a common constrain to male partner involvement in Ghana [37] even before the COVID-19 pandemic. He, 37-year-old, said;

*Now, they don't allow us (men) to be with our wives at ANC–we have to wait for them outside or stay in our cars till they are done. The workers say the place is small so due to social distancing, we should stay outside. What then is the use of coming with my partner? Most of us (men) have stopped going with our wives now. It's not good.*

## Discussion

This qualitative study explored the impact of COVID-19 on health knowledge, health-related lifestyles and health seeking behaviour among adults in the Cape Coast Metropolis. We found that the pandemic has resulted in improved health knowledge and health behaviours. Similarly, healthcare seeking behaviour has improved although many people have been negatively affected especially due to the restrictions put in place to control the spread of the COVID-19 infection.

Our study found that the pandemic has resulted in improved health knowledge among the population. This is consistent with the intensive public health education including preventive behavioural change messages being disseminated through various media (television, radio, print media, and social media) across Ghana [38, 39]. This could have resulted from the use of creative arts in translating COVID-19 information in ways that people are able to connect emotionally to create social awareness thereby, strengthening COVID-19 public health communication through improved public understanding [40]. The measures to controlling the COVID-19 pandemic in Ghana have used health education and literacy to improve access to health information [41] using mass media to establish local information networks and adapting educational messages to community beliefs and concerns [42]. In relation to the conceptual framework, health knowledge is argued to be predisposing factor which influences an individual's health behaviour [18] while its access is considered an enabling factor to behaviour

change [23]. Hence, the increases access to health knowledge due to numerous health education campaigns during the pandemic has improved individual's predisposition to positive health behaviour, understanding of health issues.

Regarding the finding that health-related lifestyles had improved among the participants, Brauer [43] posits that during pandemic including COVID-19, people change their behaviours. Assumption of positive behaviours has been argued to result from increased access to and magnitude of health informational campaigns which leads to effective and fast behavioural modifications [44, 45]. Thus, this finding supports the argument that adaptive and protective behaviour change in response to pandemic should be encourage [46] and agrees with the study by Min et al. [39] where significant improvement in food safety knowledge was observed in communities with existence of COVID-19 cases. Health education aims to provide health information and knowledge to individuals and populations and equip them with skills to be able to voluntarily adopt healthy behaviours [47]. Also, the increase in positive health behaviours and ceasing or reduction in negative health-related lifestyles could have resulted from self-preservation, a common psychological response in COVID-19 [48], resulting from improved health knowledge and risk perception. Again, the framework posits that improved health knowledge which influences attitude and perception increases the chances of individuals adopting appropriate health behaviour [18, 24] including physical exercising, healthier diet choices and ceasing of risky behaviours like alcohol consumption and smoking. Thus, the adoption of healthy behaviours and avoiding of risky behaviours is consistent with the improved health knowledge as a result of increased access to health information in this pandemic period.

Again, we observed improved healthcare seeking behaviour among some of the residents which may have resulted from improved perception of risk of exposure and perceived severity of selected health conditions such as chronic diseases, food borne diseases, and COVID-19 as well as perceived efficiency of coping or preventive strategy [49] resulting from gained information. Hence, the type and amount of information communicated to individuals and the focus on specific health information could have heightened perceived risk [50]. Their positive health seeking behaviour change, thus, results from improved understanding of health conditions and risky health behaviours. This finding is consistent with the framework that supportive predisposing factors like improved health knowledge, need for care factors such as adequate perception of health risk and severity together with enabling factors like access to health information lead to better health seeking behaviour [18, 22].

More so, our finding that some residents experienced poor healthcare seeking behaviour due to COVID-19 restrictions and protocols is in congruence to the position by Balhara et al. [51] and Yau et al. [48] that health-seeking behaviour continues to be significantly disrupted by the COVID-19 pandemic. This comes at the backdrop that outpatient and preventive care have changed significantly due to the COVID-19 pandemic with deferring of elective and preventive care visits, patients avoiding visits to reduce risk of exposure from leaving home [52, 53]. The pandemic has also resulted in untold economic and social hardship on individuals making it difficult to access health service. This finding also supports the conceptual framework which Andersen and Newman [24] posit that the presence of negative factors such as limited access to care and health resources, poor understanding of policies and poor socio-economic status hinders the decision to use health service leading to poor care seeking behaviour.

Poor health seeking behaviour has negative implications for achieving the Sustainable Development Goal (SDG) 3 of ensuring health for all at all ages through promotion of health and provision of quality healthcare services [54]. Should the poor health seeking behaviour persist, the strides made towards achieving this goal would most likely be lost.

## Strengths and limitations

The study is a novel inquiry into the significance of management of a pandemic on health-care seeking behaviour and the general health system. The study relied on verbal reports by the participants which has the potential of resulting in recall bias and overreporting or underreporting of socially acceptable and unacceptable behaviours respectively. However, participants were encouraged to be honest in their reports and guaranteed on their privacy and confidentiality of their responses while probes were used as mechanism to verify participants' views.

## Conclusion

There has been a positive impact of COVID-19 and its associated management and control measures as well as reforms on health knowledge, health-related lifestyles, and healthcare seeking behaviour among adult residents in the resource-limited setting we studied. The implication of this finding is that although increasing cases of COVID-19 will overburden the health system, efforts put in place are likely to improve health outcomes such as chronic diseases, for majority of the population. COVID-19 associated conscious and unconscious reforms could be a window of opportunity to harness, in order to improve health systems, healthcare seeking behaviour and overall health outcomes even after the pandemic wades off. Thus, health promotion and education interventions put in place should be sustained as part of the regular healthcare structure and financing. It is also important to understand the impact of reduced utilization of healthcare services, as persons with chronic diseases might succumb, not only to COVID-19 if they become infected, but also to the development of complications from pre-existing conditions.

## Author Contributions

**Conceptualization:** Farrukh Ishaque Saah, Abdul-Aziz Seidu, Luchuo Engelbert Bain.

**Data curation:** Farrukh Ishaque Saah, Luchuo Engelbert Bain.

**Formal analysis:** Farrukh Ishaque Saah.

**Funding acquisition:** Farrukh Ishaque Saah, Hubert Amu, Abdul-Aziz Seidu, Luchuo Engelbert Bain.

**Investigation:** Farrukh Ishaque Saah, Luchuo Engelbert Bain.

**Methodology:** Farrukh Ishaque Saah, Hubert Amu, Abdul-Aziz Seidu, Luchuo Engelbert Bain.

**Project administration:** Farrukh Ishaque Saah, Luchuo Engelbert Bain.

**Resources:** Farrukh Ishaque Saah, Hubert Amu, Abdul-Aziz Seidu, Luchuo Engelbert Bain.

**Software:** Farrukh Ishaque Saah.

**Supervision:** Hubert Amu, Abdul-Aziz Seidu, Luchuo Engelbert Bain.

**Visualization:** Farrukh Ishaque Saah.

**Writing – original draft:** Farrukh Ishaque Saah, Abdul-Aziz Seidu, Luchuo Engelbert Bain.

**Writing – review & editing:** Farrukh Ishaque Saah, Hubert Amu, Abdul-Aziz Seidu, Luchuo Engelbert Bain.

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
