## [Decision Letter · Decision Letter 0]

8 Apr 2021

PONE-D-21-05981

Health Knowledge and Care Seeking Behaviour in Resource-Limited Settings amidst the COVID-19 Pandemic: A qualitative study in Ghana

PLOS ONE

Dear Dr. Saah,

Thank you for submitting your manuscript to PLOS ONE. After careful consideration, we feel that it has merit but does not fully meet PLOS ONE’s publication criteria as it currently stands. Therefore, we invite you to submit a revised version of the manuscript that addresses the points raised during the review process.

Your manuscript has undergone the peer-review process and the reviewers have provided their comments/suggestions. Kindly address these points/concerns before we make a decision.

We look forward to receiving your revised manuscript.

Kind regards,

Kingston Rajiah

Academic Editor

PLOS ONE

Journal Requirements:

Please consider including more information on the number of interviewers, their training and characteristics; and please provide the interview guide used.

In your Data Availability statement, you have not specified where the minimal data set underlying the results described in your manuscript can be found. PLOS defines a study's minimal data set as the underlying data used to reach the conclusions drawn in the manuscript and any additional data required to replicate the reported study findings in their entirety. All PLOS journals require that the minimal data set be made fully available. For more information about our data policy, please see http://journals.plos.org/plosone/s/data-availability.

Reviewers' comments:

Reviewer's Responses to Questions

**Comments to the Author**

1. Is the manuscript technically sound, and do the data support the conclusions?

Reviewer #1: Yes

Reviewer #2: Yes

Reviewer #3: Yes

2. Has the statistical analysis been performed appropriately and rigorously? 

Reviewer #1: Yes

Reviewer #2: Yes

Reviewer #3: N/A

3. Have the authors made all data underlying the findings in their manuscript fully available?

Reviewer #1: Yes

Reviewer #2: Yes

Reviewer #3: Yes

4. Is the manuscript presented in an intelligible fashion and written in standard English?

Reviewer #1: Yes

Reviewer #2: Yes

Reviewer #3: Yes

5. Review Comments to the Author

Reviewer #1: This manuscript present a novel results on the effect of COVID-19 on health knowledge and care seeking behaviours in Cape Coast Metropolis. However, there are few minor issues that need t be addressed to further improve the quality of the manuscript.

INTRODUCTION

The authors should consider revising the first sentence of the second paragraph that starts with “The increasing burden of COVID-19 has resulted in various international………………..”

The first sentence of the third paragraph needs a citation. This sentence reads “The emergence of COVID–19 has deepened the strain on health systems across the globe more

especially the already overburdened……………….

METHODS

The statement that reads “The study followed the Consolidated Criteria for Reporting Qualitative Research (COREQ) guideline in reporting this study” needs to be cited.

At the study population and sampling, could the authors specify the age category for the adult they used in their study? Also, could the authors justify why they included only residents who lived for at least 6months in the study?

REFERENCES

The authors should work on the reference list. The numbers that need revision include 4, 12, 13, 15, 18, 20, 29, 34, 36, 46, and 48.

Reviewer #2: Review of “Health Knowledge and Care Seeking Behaviour in Resource-Limited Settings amidst the COVID-19 Pandemic: A qualitative study in Ghana”

Generally the entire manuscript requires some amount of editing, some of the phrases are poorly constructed, some of the sentences lack verbs, wrong phrases, wrong tenses, lack of parallel structure in a few of the sentences.

Ethics statement

• Metropolitan Health Directorate IRB (DHRCIRB/15/05/17): Is it the Dodowa Health and Research Centre IRB or the Metropolitan health directorate that approved the study? Correct this.

Abstract

Page 3: “COVID–19 associated conscious and unconscious reforms should systematically be harnessed”, need explanation or listing of exactly what authors are referring to.

Introduction

Page 5, second sentence in the last paragraph is missing “of”

Conceptual framework

• Paragraph 1: What are the three tenets? Mention them

• What are the few flaws of the model? List them and how this current study dealt or avoided them.

• What are these tenets and how are they in line with the study objectives, kindly explain

• Paragraph 1, the statement “This fit well with the study’s objective of assessing impact of the COVID-19 pandemic on the health knowledge of individuals.” Is that the entire objective of the study, I thought the model is more encompassing than making provision for only knowledge and the same with the study.

Methods and materials

• Last sentence on page 8 to first sentence on page 9: Check the sentence, the tense is not appropriate.

Study population and sampling

• Page 9, the following sentence “Only residents who had lived in the metropolis for at least six months within the period of COVID–19 pandemic in Ghana.” Says nothing, it is missing a verb. Check other paragraphs for such kinds of phrases as there are several in the entire manuscript.

• Page 9, kindly explain the following phrase “Recruitment was purposive and prospective.”

• How was saturation attained? Kindly explain.

Procedures

• Kindly complete title “procedure” does not mean anything.

• Consider improving the second sentence in the first paragraph, it is difficult to comprehend.

• Kindly explain what this means “the instrument was self-developed from literature”

• Please provide the following: age range of participants, any refusals, language that the study was carried in and how the research assistants were trained. Also, indicate how many other researchers were involved.

• Check the last sentence, I am not sure that “Collaborate” is the appropriate word, probably “corroborate”

Ethical issues

• I am not sure the study “took” approval from the Metropolitan Health Directorate following ethical approval (DHRCIRB/15/05/17) from the Dodowa Research Centre.

Correction: I think the study sought permission from the Metropolitan Health Directorate. Also, which Metropolitan Health Directorate are researchers referring to, kindly be specific. Also, correct the full name of the Dodowa Research Centre, it is not complete. Also correct it in the ethics statement and replace it with the Metropolitan Health Directorate.

Data analysis

• I don't understand how literature review is reported as a theme derived from your analysis

• Kindly explain “internal homogeneity” and “external heterogeneity”.

Results

Pages 13-19: The results section as it stands needs some improvement in analysis and write up. Researchers have resorted to putting most of the information in direct quotations, without critically analyzing the crucial messages that they contain. For instance several extensive quotes with very little explanation have been presented under all the themes. Authors should revise the results section by providing a deeper level of analysis with fewer quotes but more paraphrased and meaningful presentation.

• Opening sentence in results section: mention only main themes and link them to the framework and delete other irrelevant information that does not add anything to the introduction.

• Table 1. Structure the third major theme in the table into negative and positive effects and list the sub themes of each under is rightful theme

• Page 13, kindly explain or list “community public address systems”

• Delete “the” from sentence with the following phrase “due to their availability to majority of the Ghana’s population”

Discussion

• The authors do not refer to the conceptual framework in this section and the extent to which it helped them to answer their question or to achieve their study object. They also failed to indicate how they dealt with the flaws that were mentioned in the conceptual framework.

Reviewer #3: A couple of typographical errors and the use of articles have been noted and comments have been made in the attached document. The author(s) should be consistent in the language (Bristish or American English) choice.

6. PLOS authors have the option to publish the peer review history of their article (what does this mean?). If published, this will include your full peer review and any attached files.

Reviewer #1: No

Reviewer #2: No

Reviewer #3: No

---

## [Author Response · Author response to Decision Letter 0]

15 Apr 2021

Response to Reviewers

Reviewer #1

Introduction

Comment: The authors should consider revising the first sentence of the second paragraph that starts with “The increasing burden of COVID-19 has resulted in various international………………..”

Response: The sentence has been revised as suggested (See page 4).

Comment: The first sentence of the third paragraph needs a citation. This sentence reads “The emergence of COVID–19 has deepened the strain on health systems across the globe more especially the already overburdened……………….

Response: A citation has been provided as suggested by the reviewer (See page 4).

Methods

Comment: The statement that reads “The study followed the Consolidated Criteria for Reporting Qualitative Research (COREQ) guideline in reporting this study” needs to be cited.

Response: The statement has been cited (citation 27) (See page 9).

Comment: At the study population and sampling, could the authors specify the age category for the adult they used in their study? Also, could the authors justify why they included only residents who lived for at least 6months in the study?

Response: The study population involved adults aged 18 or older (see page 9). Also, the 6 months period begins from March, 2020 when Cape Coast Metropolis recorded first case of COVID following Ghana first recorded COVID-19 cases in March (see page 10).

References

Comment: The authors should work on the reference list. The numbers that need revision include 4, 12, 13, 15, 18, 20, 29, 34, 36, 46, and 48.

Response: Reference list has been revised and any errors corrected (See page 24).

Reviewer #2

Review of “Health Knowledge and Care Seeking Behaviour in Resource-Limited Settings amidst the COVID-19 Pandemic: A qualitative study in Ghana”. Generally, the entire manuscript requires some amount of editing, some of the phrases are poorly constructed, some of the sentences lack verbs, wrong phrases, wrong tenses, lack of parallel structure in a few of the sentences.

Ethics statement

Comment: Metropolitan Health Directorate IRB (DHRCIRB/15/05/17): Is it the Dodowa Health and Research Centre IRB or the Metropolitan health directorate that approved the study? Correct this.

Response: Approval was provided by the Cape Coast Metropolitan Health Directorate. This has thus been revised in the manuscript to remove the Dodowa Health and Research Centre which was an oversight (see page 11).

Abstract

Comment: Page 3: “COVID–19 associated conscious and unconscious reforms should systematically be harnessed”, need explanation or listing of exactly what authors are referring to.

Response: This has been explained in the concluding section of the manuscript. We felt that explaining in the abstract will cause it to exceed the expected number of words.

Introduction

Comment: Page 5, second sentence in the last paragraph is missing “of”

Response: “of” has been inserted in the second sentence of the last paragraph on page 5.

Conceptual framework

Comment; Paragraph 1: What are the three tenets? Mention them

Response: The three tenets have been stated (See page 6).

What are the few flaws of the model? List them and how this current study dealt or avoided them.

Response: This has been addressed (See page 7-8).

Comment: What are these tenets and how are they in line with the study objectives, kindly explain

Response: This has been addressed on page 6 and 8 of the manuscript.

Comment: Paragraph 1, the statement “This fit well with the study’s objective of assessing impact of the COVID-19 pandemic on the health knowledge of individuals.” Is that the entire objective of the study, I thought the model is more encompassing than making provision for only knowledge and the same with the study.

Response: This statement has been revised on page 6–7.

Methods and materials

Comment: Last sentence on page 8 to first sentence on page 9: Check the sentence, the tense is not appropriate.

Response: The sentence has been revised to correct the tense (See page 9)

Study population and sampling

Comment: Page 9, the following sentence “Only residents who had lived in the metropolis for at least six months within the period of COVID–19 pandemic in Ghana.” Says nothing, it is missing a verb. Check other paragraphs for such kinds of phrases as there are several in the entire manuscript.

Response: This sentence has been revised on page 10.

Comment: Page 9, kindly explain the following phrase “Recruitment was purposive and prospective.”

Response: The phrase “recruitment was purposive and prospective” has been revised on page 10

Comment: How was saturation attained? Kindly explain.

Response: This has been explained on page 10 of the revised manuscript

Procedures

Comment: Kindly complete title “procedure” does not mean anything.

Response: The title “procedures” has been clarified to “Study procedures” on page 10.

Comment: Consider improving the second sentence in the first paragraph, it is difficult to comprehend.

Response: The sentence has been revised to improve comprehension on page 10.

Comment: Kindly explain what this means “the instrument was self-developed from literature”

Response: the statement has been revised on page 10.

Comment: Please provide the following: age range of participants, any refusals, language that the study was carried in and how the research assistants were trained. Also, indicate how many other researchers were involved.

Response: The age range of the study participants (see page 10), languages used in the interviews and how the research assistants were trained, and other researchers involved in the data collection (page 11) have been included 

Comment: Check the last sentence, I am not sure that “Collaborate” is the appropriate word, probably “corroborate”

Response: The sentence has been revised and “collaborate” replaced with “corroborate” on page 11.

Ethical issues

Comment: I am not sure the study “took” approval from the Metropolitan Health Directorate following ethical approval (DHRCIRB/15/05/17) from the Dodowa Research Centre. Correction: I think the study sought permission from the Metropolitan Health Directorate. Also, which Metropolitan Health Directorate are researchers referring to, kindly be specific. Also, correct the full name of the Dodowa Research Centre, it is not complete. Also correct it in the ethics statement and replace it with the Metropolitan Health Directorate.

Response: The ethical statement has been revised to remove Dodowa Health and Research Centre which was an oversight. The ethical approval was given by the Cape Coast Metropolitan Health Directorate which has been indicated on page 11.

Data analysis

Comment: I don't understand how literature review is reported as a theme derived from your analysis

Response: The statement sought to explain that some of the themes identified from the data were based on literature. It has thus, been revised for clarity on page 12.

Comment: Kindly explain “internal homogeneity” and “external heterogeneity”.

Response: Internal homogeneity and external heterogeneity have been explained in parenthesis on page 12.

Results

Comment: Pages 13-19: The results section as it stands needs some improvement in analysis and write up. Researchers have resorted to putting most of the information in direct quotations, without critically analyzing the crucial messages that they contain. For instance, several extensive quotes with very little explanation have been presented under all the themes. Authors should revise the results section by providing a deeper level of analysis with fewer quotes but more paraphrased and meaningful presentation.

Response: The results section has been improved taken into consideration the concerns raised (see pages 13-21)

Comment: Opening sentence in results section: mention only main themes and link them to the framework and delete other irrelevant information that does not add anything to the introduction.

Response: This section has been revised as suggested.

Comment: Table 1. Structure the third major theme in the table into negative and positive effects and list the sub themes of each under is rightful theme

Response: The table has been restructured as suggested on page 13.

Comment: Page 13, kindly explain or list “community public address systems”

Response: This has been revised to a more self-explanatory term “community information centres” on page 14.

Comment: Delete “the” from sentence with the following phrase “due to their availability to majority of the Ghana’s population”

Response: “the” has been deleted from the sentence on page 14.

Discussion

Comment: The authors do not refer to the conceptual framework in this section and the extent to which it helped them to answer their question or to achieve their study object. They also failed to indicate how they dealt with the flaws that were mentioned in the conceptual framework.

Response: The conceptual framework has been referred to throughout the discussion pointing out how it explains the observations made (see pages 23–22).

Reviewer #3

Comment: A couple of typographical errors and the use of articles have been noted and comments have been made in the attached document. The author(s) should be consistent in the language (Bristish or American English) choice.

Response: The manuscript has been revised to conform to British English and all typographical errors have been rectified.

---

## [Editor Report · Decision Letter 1]

19 Apr 2021

Health Knowledge and Care Seeking Behaviour in Resource-Limited Settings amidst the COVID-19 Pandemic: A qualitative study in Ghana

PONE-D-21-05981R1

Dear Dr. Saah,

We’re pleased to inform you that your manuscript has been judged scientifically suitable for publication and will be formally accepted for publication once it meets all outstanding technical requirements.

Kind regards,

Kingston Rajiah

Academic Editor

PLOS ONE
---

## [Editor Report · Acceptance letter]

27 Apr 2021

PONE-D-21-05981R1 

Health Knowledge and Care Seeking Behaviour in Resource-Limited Settings amidst the COVID-19 Pandemic: A Qualitative Study in Ghana 

Dear Dr. Saah:

I'm pleased to inform you that your manuscript has been deemed suitable for publication in PLOS ONE. Congratulations! Your manuscript is now with our production department. 

Kind regards, 

on behalf of

Dr. Kingston Rajiah 

Academic Editor

PLOS ONE